

# Historical Northern Hemisphere snow cover trends and projected changes in the CMIP-6 multi-model ensemble

Lawrence Mudryk[1], Maria Santolaria-Otín[2], Gerhard Krinner[2], Martin Ménégoz[2], Chris Derksen[1], Claire Brutel-Vuilmet[2], Mike Brady[1], Richard Essery[3]

[1]Climate Research Division, Environment and Climate Change Canada, Toronto, M3H 5T4, Canada
[2]Univ. Grenoble Alpes, CNRS, IGE, 38000, Grenoble, France
[3]School of GeoSciences, University of Edinburgh, Edinburgh, EH9 3FF, UK

*Correspondence to*: Lawrence Mudryk (Lawrence.Mudryk@canada.ca)

**Abstract.** This paper presents an analysis of observed and simulated historical snow cover extent and snow mass, along with

future snow cover projections from models participating in the 6th phase of the World Climate Research Programme Coupled Model Inter-comparison Project (CMIP-6). Where appropriate, the CMIP-6 output is compared to CMIP-5 results in order to assess progress (or absence thereof) between successive model generations. An ensemble of six products are used to produce a new time series of northern hemisphere snow extent anomalies and trends; a subset of four of these products are used for snow mass. Trends in snow extent over 1981-2018 are negative in all months, and exceed -50 x $10^3$ km$^2$ during November,

December, March, and May. Snow mass trends are approximately -5 Gt/year or more for all months from December to May. Overall, the CMIP-6 multi-model ensemble better represents the snow extent climatology over the 1981-2014 period for all months, correcting a low bias in CMIP-5. Simulated snow extent and snow mass trends over the 1981-2014 period are slightly stronger in CMIP-6 than in CMIP-5, although large inter-model spread remains in the simulated trends for both variables. There is a single linear relationship between projected spring snow extent and global surface air temperature (GSAT) changes,

which is valid across all scenarios. This finding suggests that Northern Hemisphere spring snow extent will decrease by about 8% relative to the 1995-2014 level per °C of GSAT increase. The sensitivity of snow to temperature forcing largely explains the absence of any climate change pathway dependency, similar to other fast response components of the cryosphere such as sea ice and near surface permafrost.

## 1 Introduction

It is imperative that Earth System Models properly treat seasonal snow cover in order to account for a number of important energy and water cycle processes:

1. Like summer sea ice in the Arctic, spring snow cover over land has a cooling effect on the climate system (Flanner et al., 2011). The magnitude of this cooling influence has declined alongside observed reductions in spring snow cover over recent

decades (Zhang et al., 2019; Letterly et al., 2018).



2. Snow cover properties modulate the soil thermal state and thereby affect the carbon balance. Across the boreal forest, gross primary production and carbon uptake is directly related to the timing of spring snow melt (Pulliainen et al., 2017). Winter season carbon losses from northern permafrost may now be greater than the average growing season carbon uptake across tundra regions, due in part to the insulating properties of snow (Natali et al., 2019).

3. Freshwater from snow melt is a volatile natural resource, subject to inter-annual variability and trends in temperature and precipitation. Observed and projected changes to seasonal snow cover, particularly in mountain regions, have profound impacts on the quantity, quality, and timing of water availability (Fyfe et al., 2017; Marty et al., 2017; Verfaillie et al., 2018).

Observational evidence shows that snow cover is changing across mid-latitude (Mudryk et al., 2017), alpine (Hock et al., 2019), and subarctic and Arctic regions (Brown et al., 2017; Meredith et al., 2019). These changes are challenging to synthesize because of a high degree of region-to-region and season-to-season variability in trends, and because of the range of metrics by which snow can be characterized including snow extent, snow cover duration, and snow mass. While snow extent anomalies and trends are strongly related to surface temperature (Mudryk et al., 2017; Brutel-Vuilmet et al., 2013; Robinson and Dewey, 1990), snow mass changes are more challenging to diagnose because of the competing influences of temperature and precipitation, and the influence of additional processes such as sublimation through the snow accumulation season (Sospedra-Alfonso and Merryfield, 2017).

Over the past three generations of climate models (assessed through subsequent phases of the Coupled Model Inter-comparison Project – CMIP-3, CMIP-5, and CMIP-6), land surface models have increased in complexity, while moving to a finer spatial resolution. The treatment of snow within these models has grown more sophisticated, particularly with respect to sub-grid heterogeneity and snow layering, and parameterizations involving albedo, thermal conductivity, snow density, surface exchange, and snow-vegetation interactions (Essery et al., 2012; Qu and Hall, 2007). Some improvements in the simulation of snow extent and snow mass were noted between Atmospheric Model Inter-comparison Project Phase 1 (AMIP-1) and Phase 2 (AMIP-2) models (Frei et al., 1998; Frei et al., 2005). An assessment of the literature, however, shows relatively small differences in Coupled Model Inter-comparison Project Phase 3 (CMIP-3) versus Phase 5 (CMIP-5) model performance at the continental scale, and our level of understanding of how key snow processes influence model performance has improved only modestly (Menard et al., in review).

In general, CMIP-3 and CMIP-5 historical simulations show reasonable snow extent and snow mass climatologies, albeit with greater uncertainty in alpine areas (particularly western Canada, Norway, and the Tibetan Plateau/Himalaya; Brown and Mote, 2009; Brown et al., 2017). Both generations of models tend to underestimate observed spring snow extent reductions compared to observations (Frei et al., 2003; Flanner et al., 2008; Derksen and Brown, 2012). Explanations include reduced snow





sensitivity (snow loss per degree of warming; Mudryk et al., 2017), underestimated spring warming over extra-tropical land
      (Brutel-Vuilmet et al., 2013) and biases in climatological snow cover (low bias in simulated snow means there is less snow to
      lose; Thackeray et al., 2016). Projected spring snow extent loss in CMIP-5 models was primarily explained by extratropical
      temperature trends, so scenario dependence for the magnitude of snow loss emerges after 2050 (Thackeray et al., 2016;
      Meredith et al., 2019). The patterns of snow cover duration and seasonal maximum snow mass projections are similar between

CMIP-3 and CMIP-5 models (Brown and Mote, 2009; Brown et al., 2017). Snow mass is projected to increase in the coldest
      parts of the Northern Hemisphere continents and decrease across most midlatitude (Raisanen, 2008; Brown et al., 2017) and
      alpine areas (Fyfe et al., 2018; Verfaillie et al., 2018), while snow cover duration is expected to decrease everywhere (Brown
      and Mote, 2009; Brown et al., 2017). An increased frequency of low snow years emerges at 2°C global warming (Diffenbaugh
      et al., 2013).


      In part, the lack of dramatic forward progress between CMIP-3 and CMIP-5 is directly tied to continued problematic
      approaches to snow modeling. For instance, unrealistic vegetation parameters and distribution are related to errors in snow
      albedo simulations (Loranty et al. 2014; Wang et al. 2016; Thackeray et al., 2015), and some CMIP-5 models continue to use
      physically unrealistic snow albedo values (Thackeray et al., 2019). However, the choice of evaluation data may also play a

role, in that nearly all CMIP-3, and many CMIP-5 studies used a single snow product for evaluation, hence interpretation is
      prone to biases in the chosen dataset. In this study, we quantify historical trends in northern hemisphere snow extent and snow
      mass from an updated ensemble of gridded datasets. We compare these trends to historical simulations from the multi-model
      ensemble available from CMIP-6 (Eyring et al., 2016) and CMIP-5 (Taylor et al., 2012) to identify changes in model
      performance relative to historical observations. Projections of snow extent and snow mass from a range of Shared

Socioeconomic Pathways (SSPs) from the CMIP-6 ScenarioMIP (O'Neill et al., 2016) illustrate the potential range in future
      snow conditions.

## 2 Methods

### 2.1 Multi-dataset historical SCE and SWE time series

      There are numerous gridded products that utilize various combinations of surface observations, remote sensing, land surface

models, and reanalysis products to provide long time series for quantifying continental trends in snow extent and snow mass,
      and assessing climate model simulations. However, all these datasets have uncertainties related to sparse observation networks,
      satellite retrieval algorithm uncertainties, simplified model parameterizations, and/or forcing uncertainty. Overall, this means
      there is no single 'best' dataset from which variability and trends in seasonal snow can be diagnosed. Instead, the challenge is
      to combine independent datasets in a meaningful way to understand spread and uncertainty in observations and mitigate

uncertainty attributed to individual techniques (Mudryk et al., 2015; Krinner et al., 2018), much like multi-model ensembles
      and large initial condition ensembles (e.g. Kay et al., 2015) are used across the climate modeling community.



The use of multiple snow datasets (Mudryk et al., 2017; Thackeray et al., 2016; Mudryk et al., 2018) evaluated through coordinated experiments like the Satellite Snow Product Inter-comparison and Validation Experiment (SnowPEx; Mortimer et al., in discussion) represents a shift from the use of single snow datasets typical of many CMIP-3 and early CMIP-5 model evaluation studies (Brown and Mote, 2009; Derksen and Brown, 2012; Brutel-Vuilmet et al., 2013). In this study, we use six datasets that all cover the complete 1981-2018 time period. For snow extent we can extend the record back to 1967 based on one of these six component datasets, and for certain months we can further extend the record back to 1922. A summary of the datasets is provided in Table 1.

The NOAA and JASMES datasets provide a direct estimate of snow extent and were obtained from the references noted in Table 1 as total northern hemisphere (NH) snow extent time series. The four gridded snow mass datasets were regridded to 0.5° resolution, and time series were calculated for each product by integrating the volume of snow water over the NH land area, and multiplying by the density of water. Because the GlobSnow product is masked over complex topography at the native GlobSnow grid spacing of 25 km, snow mass was replaced in grid cells which contain mountains with a blend of the GlobSnow data (if any) and the mean value from the other three data sources. The weighting for the blend was determined by the mountain fraction of the grid cell area (defined using a 5 arcminute resolution topographic map as regions with a slope greater than 2 degrees). For grid cells with no mountainous terrain, unaltered GlobSnow data were used. As the fraction of mountainous terrain increases, the weight applied to the GlobSnow data is linearly reduced, reaching zero for grid cells containing only mountainous terrain.

Snow extent time series were also calculated from these four snow mass datasets by applying a threshold of 4 mm to daily snow water equivalent fields, averaging to yield monthly fractional snow cover, and summing the average monthly land area under snow. In order to merge all six snow extent datasets, the climatology and standard deviation of each data set was adjusted as follows based in part on the methodology used in Brown et al. (2010) and Brown and Robinson (2011). Each data set's climatology was replaced by the climatology of the NOAA data record, and each data set's variability was adjusted to that of the ensemble mean standard deviation. This was accomplished by first calculating standardized anomalies using each data set's own climatology and standard deviation (1981-2014) and then de-standardizing using the ensemble mean standard deviation and the climatology of the NOAA data record. The NOAA climatology was used because lacking additional verification data we assume it is the best estimate of the true historical snow extent. By contrast, we expect the variability of the 6-component dataset to be more accurate than any single dataset, and in particular, more accurate than that of the NOAA dataset alone, which may be artificially high during the spring based on several lines of evidence (e.g. Mudryk et al., 2017).

These adjusted time series were averaged over the 1981-2018 period and this average time series was merged with the adjusted NOAA time series over the 1967-1980 period. This methodology ensures that the transition between the pre- and post-1981



periods does not contain any discontinuities due to changes in climatology (e.g. were the full time series simply averaged together) or variability (e.g. were unadjusted anomalies averaged together).

Over the 1981-2018 period, 95% uncertainty bounds were determined from the standard error:

$$se = s/\sqrt{n-1} \,, \tag{1}$$

which depends on the standard deviation, s, of the n datasets included.

Over the 1967-1980 period, 95% uncertainty bounds were determined from the standard error of forecast:

$$se_f(x) = se_{res}(x) \sqrt{1 + \frac{1}{n}(1 + x^2)} \,, \tag{2}$$

where $se_f(x)$ is the standard error of the residuals from a best-fit line, $x$ is the sequence of years of the analysis period, and $n$ is the number of years of the analysis period.

For certain months, the Northern Hemisphere snow time series can be extended back to 1922 through the interpolation of a fixed network of surface snow depth observations, as described in Brown and Robinson (2011). Because of limitations in the availability of surface observations over Eurasia for during the fall and winter, the hemispheric estimates are only available for March, and April. For this longer time period, we chose the 1972-1991 reference period because we expect that the NOAA climatology over 1972-1991 is more representative of earlier 20th century snow cover (whereas snow cover will have already responded to warming temperatures by the latter portion of the record). We use the same 6-member ensemble mean standard deviation that was used to compute adjusted time series for the other data sets. This process has an implicit assumption that the variability has not changed between the early/mid 20th century period and the more recent 1981-2014 period. Uncertainty is calculated using the standard error of forecast, however we consider additional uncertainty due to the choice of reference periods used for matching the climatology and standard deviation. 95% uncertainty bounds are calculated for all possible selections of sequential 20 year climatological periods from the NOAA record in combination with all possible sequential 20 year estimates of variability. The maximum and minimum uncertainty bounds calculated from these combinations are used to define the range for each year.

Monthly snow extent trends were computed from the average of the six adjusted time series described above; monthly snow mass trends were computed from the average of the four snow mass time series.

## 2.2 Model simulations and analyses

The standard period used in this paper for evaluating the CMIP-5 (Taylor et al., 2012) and CMIP-6 (Eyring et al., 2016) model outputs is 1981 to 2014. The historical period for which simulations were driven by observed climate forcing data ends in 2014



for CMIP-6, and 2005 for CMIP-5. Observed $CO_2$ emissions between 2006 and 2014 closely follow those of the RCP8.5 emission scenario (Hayhoe et al., 2017), therefore we use the first 9 years (2006-2014) of the RCP8.5 simulations to complement the historical simulations for CMIP-5. For CMIP-6 projections, we restrict our analysis to the four Tier 1 SSPs (O'Neill et al., 2016): SSP1-2.6, SSP2-4.5, SSP3-3.7, and SSP5-8.5.


To evaluate climatological means, we analyze the first ensemble member for each model (r1i1p1 in CMIP-5 and r1i1p1f1 in CMIP-6, respectively). For trend analyses, the least squares median trend of each model's ensemble is used. No weighting is used in multi-model means. A special treatment is applied to CMIP-5, because the 1981-2014 reference period covers both the historical and the RCP8.5 scenario simulation. In order to identify the median CMIP-5 simulation over this reference period,

we concatenated the first ensemble members of the historical and the RCP8.5 scenario simulations for analyses of climatological means. For analyses of trends, we constructed a bootstrap pseudo-ensemble for each model by combining all historical and scenario ensemble members, yielding nH×nS bootstrap ensemble members for a model that had nH members in its historical ensemble and nS members in its scenario ensemble. We then identified the median snow cover and snow mass trend from this bootstrap ensemble.


In this paper, snow cover fraction [variable "snc"], snow mass ["snw"], and surface air temperature ["tas"] are used. Snow extent is not obtained from snow water equivalent by applying a threshold of 5 mm, as suggested in Brutel-Vuilmet et al. (2013). Instead, snow extent is computed directly from the snow cover fraction calculated in the models. Implications of this choice are evaluated in the discussion section.


Because we focus on seasonal snow cover, model output is masked to only take into account ice-free continental grid points. Unfortunately, not all model groups had made these time-independent model masks (land fraction ["sftlf"], ice-sheet fraction ["sftgif"]) available at the time of writing. We therefore interpolated the corresponding masks from the CNRM-CM6-1 coupled model (Voldoire et al., 2019) to the corresponding model grid. Some models accumulate permanent snow in specific areas such as

the northern parts of the Canadian Archipelago or Tibet. This means that very high snow masses can occur in the specific areas where snow does not melt entirely in summer, effectively transforming the corresponding model grid points into ice caps. This would alter our hemispheric snow mass diagnostics that are supposed to be limited to seasonal snow. We therefore limit model snow mass to a maximum of 500 kg m-2. Although this value can be exceeded for seasonal snow cover in localized regions, it should represent a reasonable maximum for snow mass at the resolution of the model simulations.


We used the CMIP-5/-6 database as available on the IPSL CICLAD computer center in December 2019. Snow cover fraction, snow water equivalent and surface temperature over the 1981–2014 period were available for twenty-three CMIP-5 models at the time of writing (see Table 2). This is a somewhat more extensive ensemble of opportunity than analyzed by Mudryk et al. (2017) and Thackeray et al. (2016). For CMIP-6 (see again Table 2), twenty-one models have these variables available over



the 1981–2014 period. Contrary to Krinner and Flanner (2018), we did not restrict the multi-model ensemble to only one model per modeling group. Because of strong similarity between multiple models emanating from a single modeling group or sharing a common development history (Masson and Knutti, 2011), this decision might result in effectively overweighting certain models that in reality only come in different flavours. By using medians of the multi-model ensemble both for climatological means and trends, we reduce possible misleading effects of this choice. The remaining Tier 1 CMIP-6 ScenarioMIP simulations

(SSP1-2.6, SSP2-4.5 and SSP3-7.0) were not used extensively in this work and are therefore not listed separately in Table 2.

## 3 Results

### 3.1 Historical trends: SCE and SWE

Because there is no established 'baseline' snow dataset with which to assess climate trends and model performance at the hemispheric scale, we use the multi-dataset approach described in Section 2.1. We argue that this approach should reduce the

potential for biased model evaluation related to uncertainty in an individual product. Previous applications have resulted in improved model verification of seasonal forecasts (Sospedra-Alfonso et al., 2016) and have produced more robust diagnosis of historical climate model simulations (e.g. Thackeray et al., 2016) compared to the use of a single dataset (Derksen and Brown, 2012).

Before evaluating the model simulations, we illustrate the monthly snow extent and snow mass trends and anomalies (1981-2018) from the multi-dataset ensemble in Figure 1. Snow extent trends are negative in all months, and exceed -50 x $10^3$ km$^2$ during November, December, March, and May. Snow mass trends are consistently negative: trends are approximately -5 Gt/year or more for all months between November and June. Both snow extent and snow mass trends are near zero during the summer because there is very little land area with snow cover.


Figure 2 shows a nearly hundred year time series of northern hemisphere snow extent, produced by combining gridded station observations (1922-1998; Brown and Robinson, 2011) available for March and April with the multi-dataset record shown in Figure 1 (described in Section 2.1). The overlap period shows good agreement between the two independent time series, albeit with greatly reduced uncertainty during the recent era. There is a high degree of inter-annual variability through the full data

record. NH snow extent peaked over the 1950 to 1970 period (particularly evident in March), with reductions since 1980 consistent with the combined datasets shown in Figure 1.

### 3.2 Historical simulations

Examining the historical model output, Figure 3 indicates that overall, the CMIP-6 multi-model ensemble better represents the snow extent climatology over the 1981-2014 period for all months (Figure 3b), correcting a low bias in CMIP-5 relative to the

multi-dataset observational ensemble (Figure 3a). One model family (EC-Earth3 and EC-Earth3-Veg, Wyser et al., 2019) is a

clear outlier due to the simulation of very high snow cover extent (dots above the box and whiskers in Figure 3b). These climatologies are statistical outliers from January to October; while not statistical outliers during November and December they are still at the origin of the large spread towards high values. This high snow extent is linked to very high snow cover fractions even for low snow mass values (see Section 4.2). If this obvious outlier is not considered, the inter-model spread is

lower for CMIP-6 compared to CMIP-5.

The higher climatological snow extent in CMIP-6 models relative to CMIP-5 might in part be due to a climatological snow mass increase relative to CMIP-5 (Figure 4). Hemispheric scale snow extent and snow mass are not always correlated in observational datasets (e.g. Ge and Gong, 2008). Higher simulated snow mass, however, tends to lead to higher simulated

snow extent due to the connected parameterizations of these variables (e.g. snow cover fraction increases with snow mass until a critical threshold when complete snow cover occurs on the grid cell scale). The reduced spread in the lower part of the simulated snow mass range (third and fourth quartiles) in CMIP-6 leads to a more significant high bias in the more recent CMIP exercise (in the sense of a reduced overlap of the inter-model distribution with the interannual range of observed NH snow mass). Correspondingly, the upper part of the range (first and second quartiles) of the CMIP-6 ensemble feature models

with higher biases than seen in CMIP-5. An important consideration, however, is that the observational snow mass ensemble is likely biased low due to underestimation of snow mass in high elevation areas (Wrzesien et al., 2018; see Section 4.1 for a more complete discussion of this issue).

Differences in the geographic patterns of CMIP-5 and CMIP-6 spring and fall snow cover (Figure 5) reflect the overall higher

CMIP-6 snow extent evident in Figure 3. The proportion of models simulating more than 50% snow cover in spring and autumn increases along the southern border of the snow-covered regions both in Northern Eurasia and North America. The increase is particularly strong and widespread in autumn (Figures 5c and 5d), resulting in the reduced CMIP-6 bias from October onwards. CMIP-6 also has an increased fraction of models simulating overly extensive snow cover in Eurasia and North America during both seasons consistent with the outlier models already mentioned.


There is a notable increase in the simulated spring snow cover across the Tibetan plateau in CMIP-6, which strengthens a regional positive bias already present in CMIP-5 simulations (Figures 5e and 5f). More than 20% of CMIP-6 models simulate persistent summer snow cover on the Tibet Plateau, and more than half of the models simulate persistent snow cover in excess of 50% areal fraction in the Hindu-Kush area further west. In South America (not shown), the CMIP-6 ensemble simulates a

median annual maximum snow extent of about 295,000 km$^2$ (mean of approximately 400,000 km$^2$, heavily influenced by some outliers), which is in good agreement with the average 1979-2006 annual maximum extent of about 320,000 km$^2$ estimated from satellite passive microwave data (Foster et al., 2009).



In summary, while some regional differences are evident, the improved hemispheric statistics seen in Figures 3 and 4 are
generally consistent with the geographic patterns shown in Figure 5. While there is some apparent skill degradation in some
areas such as the Tibet Plateau, there is a clear improvement in the mean model agreement with observations at the hemispheric
scale, particularly in October-November, that is not due to compensation of biases in different regions.

Simulated snow extent and snow mass trends over the 1981-2014 period (Figure 6) are somewhat stronger in CMIP-6 than in
CMIP-5 with respect to the inter-model median (note that the observed trends are computed over the 1981-2014 period for
consistency with the CMIP-6 ensemble and thus slightly different from the 1981-2017/18 trends given Figure 1). The observed
snow extent and snow mass trends generally fall within the model range, although this is expected given the very large inter-
model spread in the simulated trends for both variables. Improvement from CMIP-5 to CMIP-6 can be observed with respect
to fall snow cover trends, which are more strongly negative in CMIP-6 models, in closer agreement with observations. This
change is not simply due to the increased snow extent (Figure 3) because the CMIP-6 trends are approximately twice as strong
as CMIP-5, while the increase in climatological snow extent was more modest. During spring, CMIP-6 models reasonably
simulate the observed snow cover decline. CMIP-6 snow mass trends exhibit a narrower range than CMIP-5 models during
winter and spring, and are in closer agreement with observations, particularly in April and May. Observed and simulated snow
mass trends in the fall are weak because of the generally shallow snowpack during the early accumulation season. Trends for
July through September can be discounted due to the very low snow cover during these months.

In order to determine whether the 34-year trends calculated above are representative of an anthropogenic forcing signal, CMIP-
6 ensemble-mean values for monthly NH snow extent trends were computed over periods of various lengths (starting in 1981)
from 5 to 34 years as shown in Figure 7a. When considering very short periods (5 years), the ensemble-mean monthly trends
span random values that can be either positive or negative. Over longer periods, these trends all converge toward negative
values with magnitudes that are stronger during the fall and spring seasons. The trends all become more positive after the tenth
year (1991), following the eruption of Mount Pinatubo, an indication of the response of NH snow extent to the subsequent
tropospheric cooling forced by the eruption. The probability of obtaining a negative trend is shown in Figure 7b, estimated
from the range of trends in the 178-member ensemble. After five years, this probability is close to 0.5 (roughly equal numbers
of positive and negative trends found within the ensemble), highlighting the influence of internal climate variability on snow
extent at these short timescales. After 20 years, this probability ranges between 0.7 and 0.9, with lower values in spring
compared to fall, suggesting a higher interannual variability in spring than in fall. Finally, after 30 years the probability of
obtaining a negative trend is larger than 95% for all months. This result indicates that requiring a minimum of 30 years would
be justified in order to confidently differentiate anthropogenically-driven snow extent trends from internal variability.





### 3.3 Model projections


The projected CMIP-6 snow cover extent and snow mass changes shown in Figure 8 share the broad features of the CMIP-5 projections (e.g. Brutel-Vuilmet et al., 2013; not shown here). The different scenarios start to diverge in about 2040, approximately 25 years after the start of the projection period. Under SSP1-2.6, stabilization of the multi-model mean snow cover extent occurs around 2060 at about -18% ( ±7%) relative to the 1995-2014 average in spring (Figure 8a), and at about -

20% (±7%) in autumn (Figure 8b). Unabated greenhouse gas emissions underlying SSP5-8.5 lead to continuous and ongoing snow-cover losses which reach -55% (±10%) in spring and -60% (±10%) in autumn by the end of the 21$^{st}$ century. The stabilized loss of snow cover by mid-century under SSP1-2.6 compared to continued reductions to end of century under SSP5-8.5 is consistent with CMIP-5 simulations under RCP2.6 versus RCP8.5 (Meredith et al., 2019).

Projections of March snow mass (which captures the approximate timing of maximum NH snow mass) evolves similarly to snow extent, with somewhat lower losses relative to 1995-2014 (-40% ±10% by end of century in SSP5-8.5). Similar to snow cover extent, March snow mass reductions plateau after mid-century in SSP1-2.6, but continue to decline under all other SSPs. This overall reduction in peak NH snow mass will cause notable water cycle impacts (e.g. Diffenbaugh et al., 2013; Berghuijs et al., 2014). A more detailed analysis is required to confirm whether snow mass is projected to increase for the same high

latitude regions and scenarios as simulated by CMIP-5 models (e.g. Eurasian and North American Arctic under RCP8.5; see Brown et al., 2017).

The difference in snow cover evolution projected for the various greenhouse gas concentration pathways can be distilled down to a rather simple relationship when the projected snow extent is plotted as a function of the global-mean surface air

temperature (GSAT), as shown in Figure 9 for projected spring snow extent changes under four scenarios. There is a single linear relationship between spring snow extent and GSAT changes valid across all scenarios. This finding suggests that Northern Hemisphere spring snow extent will decrease by about 8% relative to the 1995-2014 level per °C of GSAT increase, which is somewhat weaker than the currently observed sensitivity (Mudryk et al., 2017). We also note that there is no significant difference in this linear relationship between CMIP-6 models with high equilibrium climate sensitivity and those

with low equilibrium climate sensitivity (not shown).

Given this single linear relationship for all scenarios, we thus restrict analysis to SSP5-8.5. This scenario leads to the strongest GSAT changes over the 21st century and thereby allows exploration of the sensitivity of snow cover to GSAT across the largest range of projected temperature changes. Figure 10a demonstrates that the snow-cover reductions projected as a function

of GSAT change are nearly linear for all months in the multi-model mean over the entire SSP5-8.5 temperature projection range, except for summer months (July to September), for which linearity necessarily breaks down when the land surface is





essentially snow-free. The increase of the snow-free season length with GSAT in the entire hemisphere is particularly visible in Figure 10b.

## 4 Discussion

### 4.1 Observed Changes

Strong snow extent trends are apparent over the 1981-2018 period based on an ensemble of six observation-based products. An extended dataset covering 1922 to 2018 provides no evidence of previous multi-decadal periods of sustained loss of snow cover during March and April comparable to that observed in recent decades. This loss of snow cover is consistent with numerous studies over the past decade which have documented the emergence of dramatic snow cover extent reductions (e.g.
Allchin and Dery, 2019; Hernández-Henríquez et al., 2015; Dery and Brown, 2007). Historical trends in spring snow extent are strongly associated with extratropical surface temperature warming (Mudryk et al., 2017; Thackeray et al., 2016). The strong negative snow extent trends during October and November in Figure 1 are consistent with dramatically warmer surface temperatures across the Arctic during fall driven by summer sea ice loss (Serreze et al., 2009).

The six dataset ensemble used in this study provides clear evidence that fall snow extent reductions are actually greater than the spring season. However, negative fall snow extent trends have not been consistently reported previously in the literature, with some studies even reporting a snow extent increase during October (e.g. Cohen et al., 2014). While the NOAA snow chart data record continues to show a positive trend in October, it is not replicated by other datasets (Brown and Derksen, 2013; Mudryk et al., 2014; Mudryk et al., 2017), including the recent Hori et al. (2017) dataset based on optical satellite data.
Increased snow extent in October remains physically unrealistic given the concurrent surface temperature trends (Mudryk et al., 2017). While the NOAA dataset remains an important component of the observational ensemble used in this study, it provides a cautionary tale against relying on single datasets to establish a climate baseline.

Winter season snow extent trends are influenced by both temperature and precipitation. While events such as cold air outbreaks
related to the behaviour of the stratospheric polar vortex can have a notable impact on continental-scale snow cover during the winter, the negative winter snow extent trends in Figure 1 indicate that warming temperatures across the marginal snow cover zone have resulted in less snow extent (and snow mass). Complete diagnosis of the snow mass trend drivers requires coordinated analysis of temperature and precipitation datasets (e.g. Sospedra-Alfonso et al., 2015), which is beyond the scope of the current study.

An ensemble of four datasets indicate reductions in total northern hemisphere snow mass over the 1981-2018 period. While we consider this mean trend robust, it is likely that the historical snow mass climatology derived from the gridded products used in this study is biased low. This low bias is expected because gridded products underestimate snow mass in alpine areas

due to coarse grid cell resolution and uncertainty in precipitation forcing across these regions (Wrzesien et al., 2018). While
this may impact the climatology used for historical model evaluation in Figure 4, the same resolution related bias also
influences climate models in mountain regions (e.g. Fyfe et al., 2017), which have an even coarser resolution than the gridded
products. In this sense, we expect climatological biases relative to 'reality' to be in the same direction for both observations
and models.

Further work is needed to explore some provocative hydrological impacts of changing snow cover, which has emerged from
recent analysis. For example, observations from western North America show that snow melt is initiated earlier in spring due
to warming surface temperatures, but the subsequent melt rate is lower, which impacts the fraction of meltwater volume
produced at high snowmelt rates (Musselman et al, 2017).

### 4.2 Historical Simulations

Successive editions of the CMIP exercise provide an excellent opportunity to evaluate progress in the realism of successive
generations of climate models. In this study, we focused on the most fundamental hemispheric-scale snow metrics: snow extent
and snow mass. Concerning snow extent, a low bias during most months in CMIP-5 is largely corrected in CMIP-6, even
though some models simulate excessive snow cover extent for all months (Figure 3). Hemispheric snow mass is overestimated
both in CMIP-5 and CMIP-6 (Figure 4), which warrants an analysis of the relationship between snow cover and snow mass at
the continental and grid point scales.

At the hemispheric scale, the main feature of the observed month-to-month relationship between NH snow extent and snow
mass (Figure 11) is a hysteresis arising from (1) a steep snow extent increase in autumn at relatively low total snow mass
(corresponding to a thin snowpack but expanding snow extent) and (2) quickly decreasing spring snow mass coincident with
snow line retreat during the melt season. CMIP-6 models necessarily capture this essential dynamic of global snow cover at
the seasonal scale because the annual snow cycle begins and ends with an essentially snow free NH. However, nearly all
models increase snow extent too slowly given increases in snow mass during the build up to peak snow mass, and decrease
the snow extent too slowly during the snow melt period (the slope for the model simulations falls below observations in Figure
11). Also, nearly all models overestimate the peak snow mass (as already shown in Figure 3). Only two model families (the
ECMWF and Hadley Centre models) systematically overestimate snow extent for a given total snow mass.

To shed light on why most models incorrectly represent the observed snow extent versus snow mass relationship during the
snow onset season (upper branch of the hysteresis in Figure 11), we show the relationship between snow cover fraction and
snow mass at the model grid point scale for selected models and observations during November in Figure 12. The heat map
for observations (Figure 12a) shows a rapid increase of snow cover fraction for low values of snow mass, attaining 100% at
about 10 kg.m$^{-2}$. EC-Earth3 (Wyser et al., 2019), which is one of the models exhibiting too steep a hemispheric snow extent



increase in autumn (Figure 12b), clearly overestimates the grid-point-scale snow cover fraction at low values of snow mass, reaching 100% almost immediately. Conversely, CESM2 (e.g., Gettelmann et al., 2019) is one of the models exhibiting too weak a slope in the snow extent vs. snow mass relationship on the hemispheric scale (Figure 11). Consistently, it seems to

attain 100% snow cover too late at the grid point scale (at about 30 to 40 kg.m$^{-2}$; Figure 13d), although the parameterization implemented in this model is observationally based (Niu and Yang, 2007). MIROC6 (Tatebe et al., 2019), which uses a relationship described by Nitta et al. (2014), closely follows the observed relationships both at the hemispheric (Figure 11) and grid-point (Figure 12a) scales, as seen in Figure 12c. The analysis at the grid-point scale effectively reveals the model parameterizations linking the (usually diagnosed) snow cover fraction to the prognostic snow mass. Based on this analysis, it

is tempting to recommend implementing a functional relationship that follows the maximum of the observed distribution. For example, a simple linear relationship attaining 100% at 20 kg.m$^{-2}$ might be a satisfactory representation at the hemispheric scale; however, for representation of snow cover fraction in mountainous areas, more sophisticated parameterizations are likely necessary (e.g. Helbig et al, 2015).

Excessive snow mass at the hemispheric scale is another feature of the CMIP-6 models (Figures 4 and 11), although further work is also needed to clarify the potential impact of a low snow mass bias across mountain regions in the gridded products used for evaluation. Several reasons are possible for this overestimate: an underestimate of snowpack sublimation, underestimated peak snow masses in the observationally-based datasets, excessive simulated solid precipitation rates, insufficient melt of early season snow, or some combination of these issues. Excessive solid precipitation rates have been

reported as the main reason for an overestimate of snow cover and snow depth in an intercomparison of reanalyses on the Tibetan Plateau (Orsolini et al., 2019). It might also be the reason for the strong positive snow cover and snow duration bias seen in the CMIP-6 ensemble in that region, however, this hypothesis requires further focused analyses. Compared to Global Precipitation Climatology Project data (GPCP; Adler et al., 2003) winter season (November to March) 1981-2014 precipitation estimates for the Northern Hemisphere high-latitude (>50°N) ice-free land areas, there is no apparent overestimate by the

coupled models. On average, the models even seem to underestimate the winter precipitation rates. The median CMIP-6 value is about 93% (range 65-104%) of the GPCP precipitation rate for these months. Although precipitation climatologies are known to be problematic in high latitudes and GPCP observations were reported to be biased high in Eurasia (Behrangi et al., 2016), a high precipitation bias seems unlikely to be the reason for the excessive snow mass in most CMIP-6 models. A full analysis of snow mass budget terms (for example as carried out in Sospedra-Alfonso et al., 2016) is required to fully address

this issue. Historical offline simulations from the 'Land-Hist' experiment (part of the CMIP-6 Land Surface, Snow and Soil Moisture Model Intercomparison Project -LS3MIP; van den Hurk et al., 2016) provide a useful resource for this future analysis.

## 4.3 Projections

The Earth System is highly complex and non-linear in many aspects. Northern Hemisphere snow cover properties depend on temperature and precipitation over the snow accumulation and ablation seasons, but snow melt, for example, is also heavily





determined by available solar radiation at the surface, and the incoming top-of-the-atmosphere solar radiation that this depends

on is not a function of anthropogenic climate change. Surface characteristics such as vegetation and topography vary regionally

and latitudinally, which could be expected to induce differential responses of snow cover to climate change. Furthermore,

regional climate change (temperature, precipitation) is not necessarily linearly related to GSAT change. All these complexities

make it *a priori* surprising that the projected snow extent could essentially be proportional to future GSAT changes. The CMIP-

6 ensemble, however, strongly suggests that, on a hemispheric scale, future snow extent changes can be rather unambiguously

related to GSAT, the fundamental metric of future global climate change (Figure 8).

Concerning other cryospheric elements of the climate system, similar linear relationships have been reported for sea ice (Notz

and Stroeve, 2016) and near-surface (top 2m) permafrost (Burke et al., in preparation). A common characteristic of sea ice,

near-surface permafrost and seasonal snow is that they respond quickly (in a few years or less, down to monthly timescales

for snow) to surface climate conditions. This fast response largely explains the absence of any climate change pathway

dependency. However, the near-linearity of the snow cover response, which only breaks down once almost no snow is left,

remains remarkable. It clearly links future large-scale snow cover changes to global temperature changes, and thus, via a linear

relationship between global temperature changes and cumulative $CO_2$ emissions (Collins et al., 2013), to humanity's future

socio-economic choices. However, such a clear link on the hemispheric scale does not necessarily imply that snow cover

changes at regional to watershed scales can be predicted in a similarly straightforward way. It remains a challenge to produce

trustworthy projections of snow cover changes at the scale required for the management of water resources and ecosystems.

In this respect, regional and even hemisphere-scale biases in the simulated current-day snow extent and mass, as shown in

Figures 3-6, clearly warrant continued caution in the interpretation and usage of climate model output.

**5 Conclusions**

This paper presented an evaluation of first-order characteristics of Northern Hemisphere snow cover by CMIP-6 models against

an updated ensemble of gridded snow products (six for snow extent; four for snow mass). This updated dataset reveals strong

negative NH snow extent trends in early winter and spring, and consistently strong negative snow mass trends in spring. The

spring snow extent reductions have continued unabated over the past fifteen years, and are strongly associated with

extratropical temperature trends. The multi-dataset analysis in this paper provides robust evidence of a similar decline during

the snow onset season. Because NH land areas are essentially snow free through the summer period, warming temperatures in

the fall effectively delay the snowfall events required to initiate the snow cover season and advance the snowline. Snow mass

trends are negative through the entire snow season, rapidly increasing in magnitude during the fall to remain and remaining

greater than -5 Gt per year through May. Snow mass trend attribution is more complicated than snow cover extent since it is

influenced by cumulative drivers over the entire accumulation season, including temperature, precipitation, and sublimation.

Compared to CMIP-5, the more recent CMIP-6 ensemble better represents the recent (1981-2014) observed mean monthly NH snow extent. The increased climatological NH snow extent in CMIP-6 should have impacts on the simulated radiative forcing and surface albedo feedback, particularly in spring when solar radiation fluxes are rapidly increasing and snow cover
extent is still high; however, analysis of these effects is beyond the scope of this article.

The NH snow mass is biased high both in CMIP-5 and CMIP-6, but because of the reduced spread in the CMIP-6 ensemble (as available at the time of writing), the high bias appears more significant in CMIP-6 than in CMIP-5. The contradiction between a correctly simulated snow extent and an overestimated snow mass seems at least in part to be explained by skewed
parameterized relationships between grid-point snow mass and snow cover fraction in many models. Further work is needed on the observational side to better constrain uncertainty in the determination of snow mass model bias.

The general characteristics of the monthly snow extent trend series are more realistic in CMIP-6; notably the strong snow cover trend in early winter is better reproduced in CMIP-6 than in the CMIP-5 ensemble. Monthly NH snow mass trends are
fairly realistic in CMIP-6, but the observed trends cover a very large uncertainty range, so it is difficult to determine if there is any notable improvement since CMIP-5.

NH snow extent exhibits a linear response to GSAT changes for all months, only interrupted at higher warming levels when NH snow cover completely vanishes during some months. This remarkable linear relationship shows no emission pathway
dependency. In all scenarios, a given GSAT change can be, within uncertainties, related to the same NH snow extent change of about $-2.0 \pm 0.1 \times 10^6$ km$^2$/°C for the winter months.

The analyses presented here will most certainly be followed by more detailed focus on snow-related characteristics of CMIP-6 output, for example concerning attribution of observed changes to human activity, snow-albedo feedback intensity, the role
of snow cover in regional climate change, and others. In particular the existence of the new CMIP-6 land-only simulations, carried out within LS3MIP (van den Hurk et al., 2016), will allow for a consistent analysis of snow-related processes using a hierarchy of simulations of increasing complexity. We hope that analysis of snow-related biases in this suite of simulations will help in attributing model deficiencies to shortcomings in either the land models, the driving atmospheric models, or biases in the simulated coupled land-atmosphere-ocean system.

**Data Availability**

The multi-dataset historical snow extent and snow water equivalent time series described in Section 2.1 will be made available on a Canadian government website. [This process is underway; currently they are provided for reviewers as supplementary



material]. The model data used in this analysis is publically available on the Earth System Grid Federation
(https://esgf.llnl.gov/).

**Author Contributions**

LM, MS, GK, and MM performed analysis and produced figures. LM, GK, and CD wrote the original draft. LM constructed
the observation-based time series. CB and MB obtained model data. All authors contributed to manuscript review and editing.

**Competing Interests**

The authors declare that they have no conflict of interest.

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



Table 1. Snow products combined to produce multi-dataset snow cover extent and snow mass datasets.

| Dataset | Time Period | Variable | Method | Reference |
|---|---|---|---|---|
| NOAA Snow Chart Climate Data Record | 1967-2018 | Snow extent | Manual analysis of primarily optical satellite imagery | Estilow et al., 2015 Robinson et al. 2012 |
| JAXA JASMES | 1981-2018 | Snow extent | Objective analysis of AVHRR imagery | Hori et al., 2017 www.eorc.jaxa.jp/cgi-bin/jasmes/monthly/jasmes_main_v3r3.cgi?area=GL&lang=en |
| Crocus | 1981-2018 | Snow mass | Crocus physical snow model driven by ERA-interim reanalysis | Brun et al., 2013 |
| MERRA-2 | 1981-2018 | Snow mass | Reanalysis (Catchment land surface model) | Gelaro et al., 2017 GMAO, 2017b |
| GlobSnow | 1981-2018 | Snow mass | Satellite passive microwave data and surface snow depth observations | Takala et al., 2011 www.globsnow.info |
| Brown | 1981-2018 | Snow mass | Simple snow model driven by ERA-interim reanalysis | Brown et al., 2003 |
| Brown and Robinson | 1922-1992 | Snow extent | Surface snow depth observations, NOAA snow charts, passive microwave remote sensing | Brown and Robinson, 2011 |




Table 2. Number of CMIP-5 and CMIP-6 model realizations with available snow cover fraction ('snc'), snow water equivalent ('snw'), and surface air temperature ('tas') for both historical and RCP 8.5/SSP5-8.5 experiments.

| CMIP-5 Model | Historical | RCP8.5 | CMIP-6 Model | Historical | SSP5-8.5 |
|---|---|---|---|---|---|
| BCC-CSM1.1 | 3 | 1 | BCC-CSM2-MR | 3 | 1 |
| BNU-ESM | 1 | 1 | | | |
| CanESM2 | 5 | 5 | CanESM5 | 25 | 25 |
| CMCC-CM | 1 | 1 | | | |
| CMCC-CMS | 1 | 1 | | | |
| CCSM4 | 9 | 6 | | | |
| CESM1-BGC | 1 | 1 | CESM2 | 11 | 2 |
| CESM1-CAM5 | 3 | 3 | | | |
| CESM1-WACCM | 4 | 3 | CESM2-WACCM | 3 | 1 |
| CNRM-CM5 | 10 | 5 | CNRM-CM6-1 | 18 | 6 |
| | | | CNRM-ESM2-1 | 5 | 5 |
| CSIRO-Mk3.6.0 | 10 | 10 | | | |
| | | | EC-Earth3 | 24 | 7 |
| | | | EC-Earth3-Veg | 4 | 3 |
| FGOALS-g2 | 5 | 1 | FGOALS-f3-L | 3 | 1 |
| | | | GFDL-ESM4 | 1 | 1 |
| | | | GFDL-CM4 | 1 | 1 |
| GISS-E2-H | 6 | 2 | GISS-E2-1-G | 10 | 10 |
| GISS-E2-R | 6 | 2 | GISS-E2-1-H | 10 | 10 |
| | | | HadGEM3-GC31-LL | 4 | 4 |
| INM-CM4 | 1 | 1 | | | |
| | | | IPSL-CM6A-LR | 32 | 6 |
| MIROC5 | 5 | 5 | MIROC6 | 10 | 3 |





| | | | | | |
|---|---|---|---|---|---|
| MIROC-ESM | 3 | 1 | MIROC-ES2L | 3 | 1 |
| MIROC-ESM-CHEM | 1 | 1 | | | |
| MPI-ESM-MR | 3 | 1 | MPI-ESM1-2-HR | 10 | 1 |
| MRI-CGCM3 | 3 | 1 | | | |
| MRI-ESM1 | 1 | 1 | MRI-ESM2-0 | 5 | 1 |
| NorESM1-M | 3 | 1 | NorESM2-LM | 3 | 1 |
| NorESM1-ME | 1 | 1 | | | |
| | | | UKESM1-0-LL | 9 | 5 |



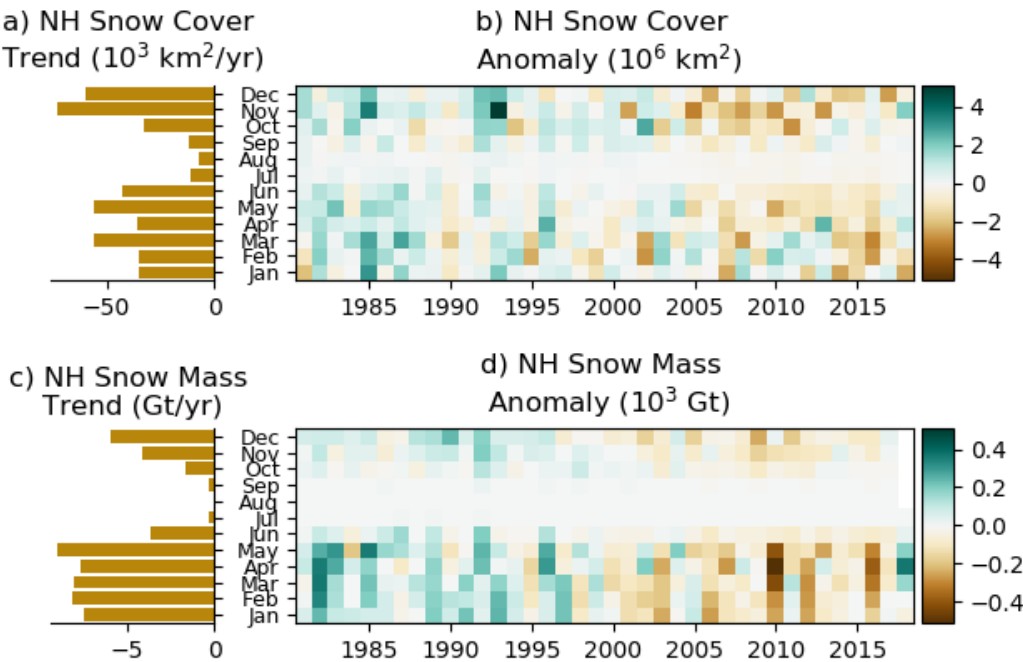

Figure 1: Observation-based NH snow extent (top: a,b) and snow mass (bottom: c,d) trends (left: a,c) and anomalies (right: b,d) for
Jan 1981 through December 2018 relative to 1981-2014.

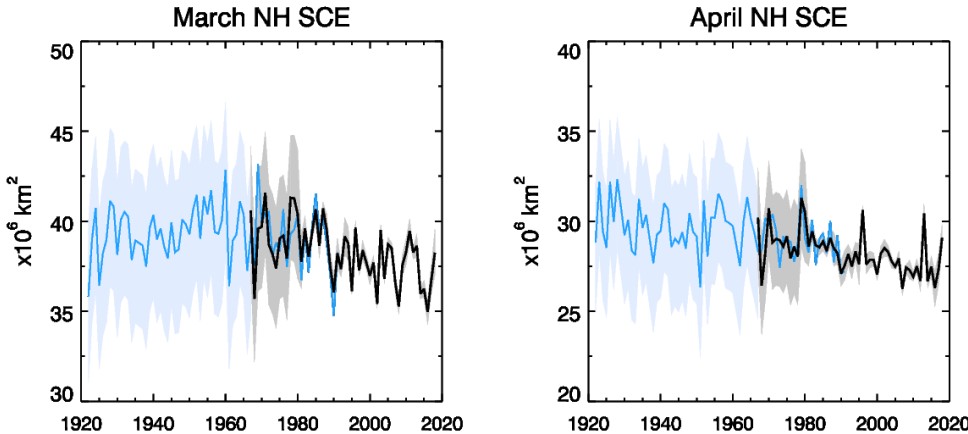

Figure 2: Northern Hemisphere snow extent spanning 1922-2018 for March (left) and April (right). Snow extent estimates (solid)
and uncertainty (shading) are based on station observations (blue) over the 1922-1991 period (shading stops at 1967) and the multi-
dataset record (black) over the 1967-2018 period.





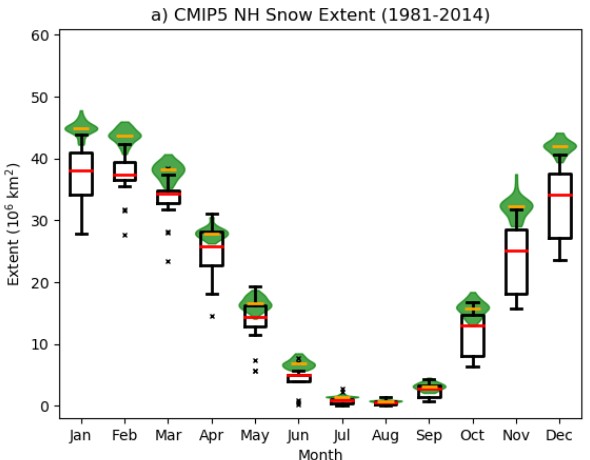
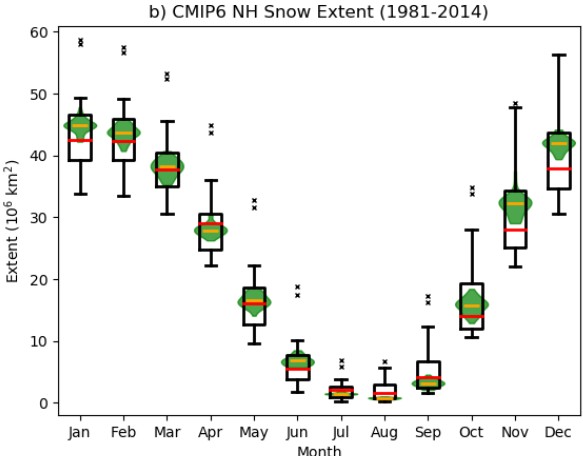

**Figure 3: Simulated CMIP-5 (a) and CMIP-6 (b) monthly mean 1981-2014 NH snow extent. The green violins (yellow bar: median year) represent the observed interannual distribution. The boxes and whiskers represent the individual models' (first ensemble member) 1981-2014 average from the CMIP-5/6 multi-model ensembles (red bar: median model).**


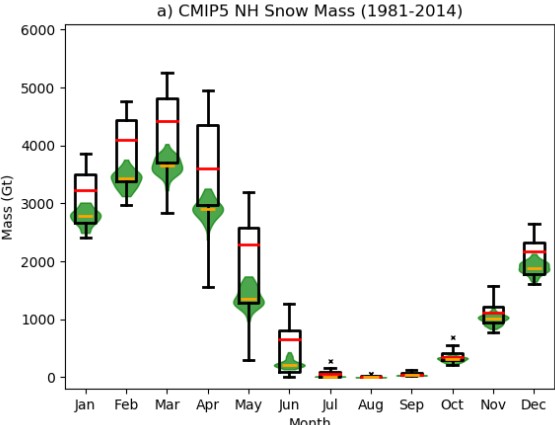
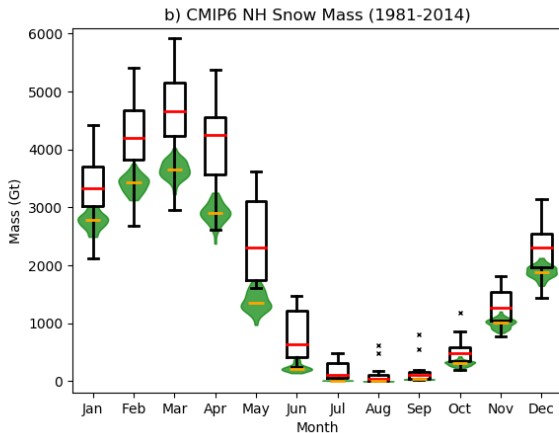

**Figure 4: As in Figure 3 but for snow mass.**





**Figure 5: Percentage of models simulating more than 50% snow cover in the Northern Hemisphere during March-April (top) and October-November (middle) for 1981-2014, for CMIP-5 (a,d) and CMIP-6(b,e). The 50% snow cover frequency line from the NOAA data for the same months during that period is shown in pink. Percentage of models simulating more than 50% snow cover in High Mountain Asia during July-August for 1981-2014, for CMIP-5 (g) and CMIP-6 (h). Differences (CMIP-6 minus CMIP-5) are shown in panels c, f, and i.**




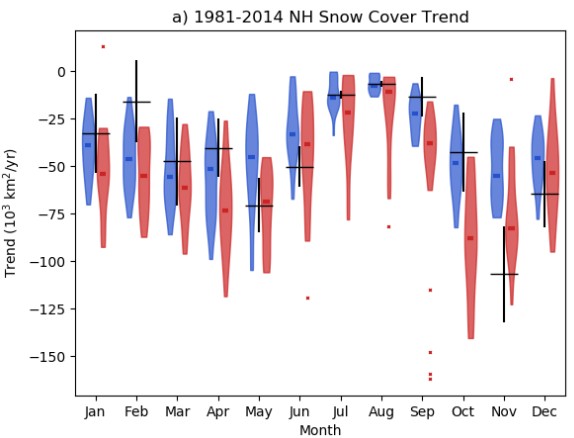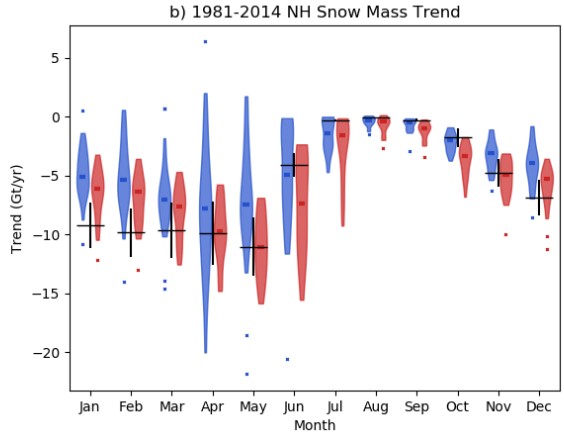

**Figure 6: Northern Hemisphere trends of snow extent (a) and snow mass (b) between 1981 and 2014, for CMIP-5 (blue) and CMIP-6 (red). The violins span the model range after the removal of outliers. The observed trends are represented by the black crosses, the height of the cross representing the 1σ uncertainty of the observed trend.**

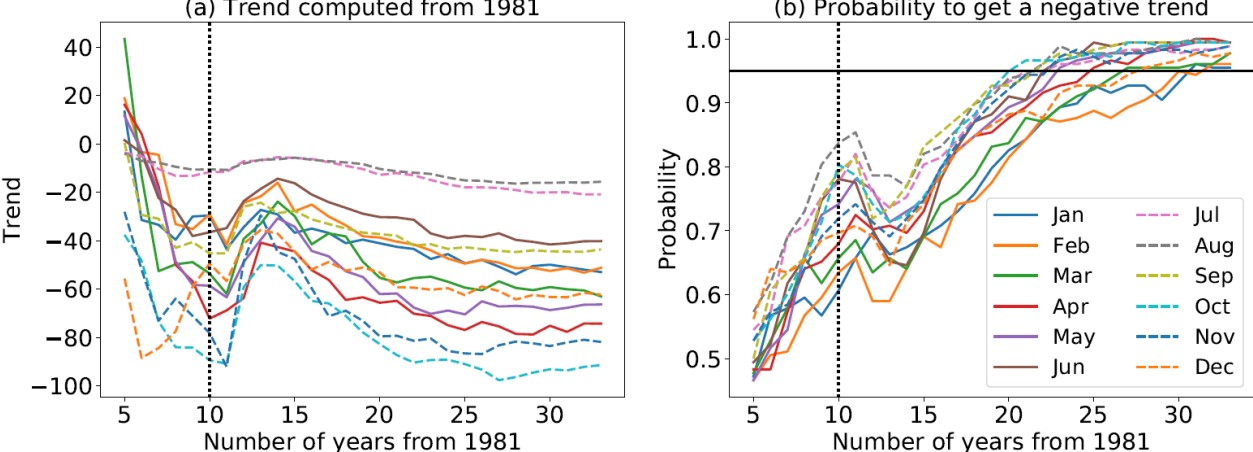

**Figure 7: (a) Northern Hemisphere monthly trends of snow extent in CMIP-6 models over 1981-2014 computed with 178 members from 21 CMIP-6 models as a function of the number of years from 1981 and starting from a 5-year time series; (b) Probability of negative monthly trends estimated from the full ensemble members. The horizontal bar highlights a probability of 95%, and the vertical bars correspond to the year of the Pinatubo eruption.**





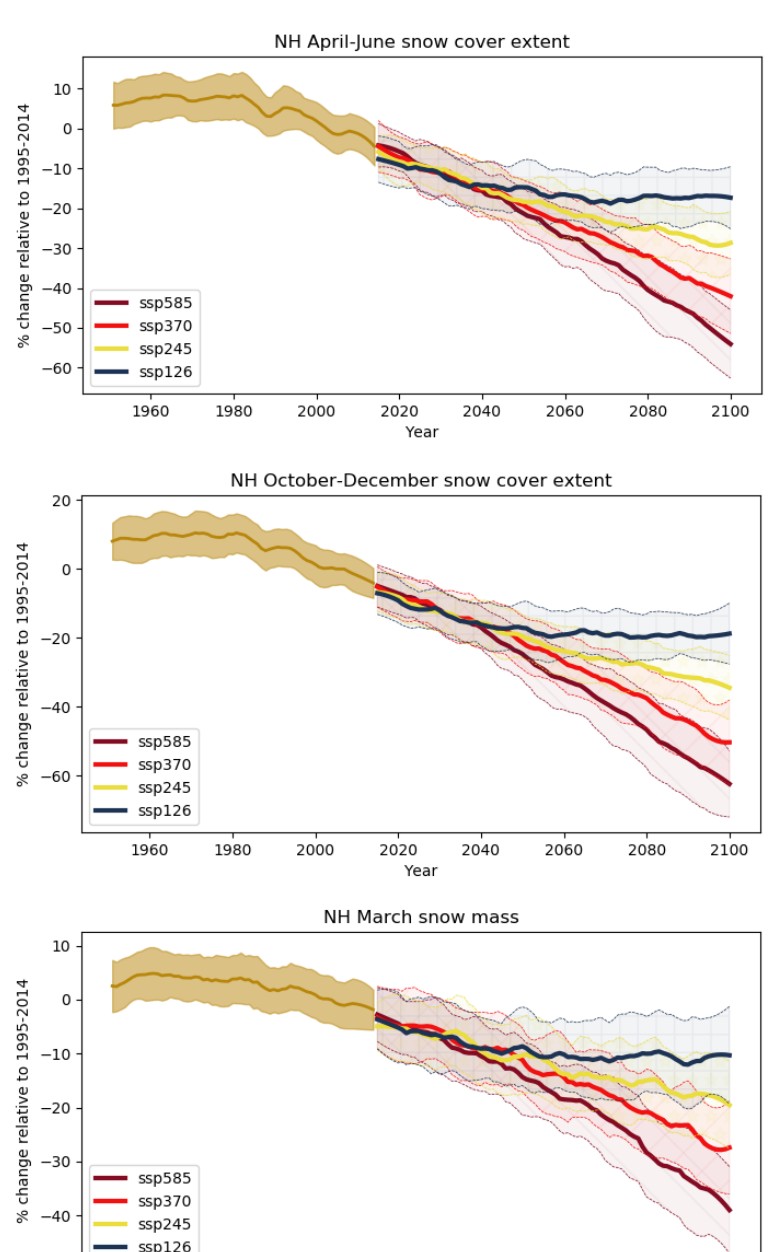


**Figure 8: Time series of NH spring (April-June) and autumn (October-December) snow cover extent and March snow mass changes (relative to the 1995-2014 average); multi-model mean of the first ensemble members of the historical and scenario runs with inter-model standard deviation.**

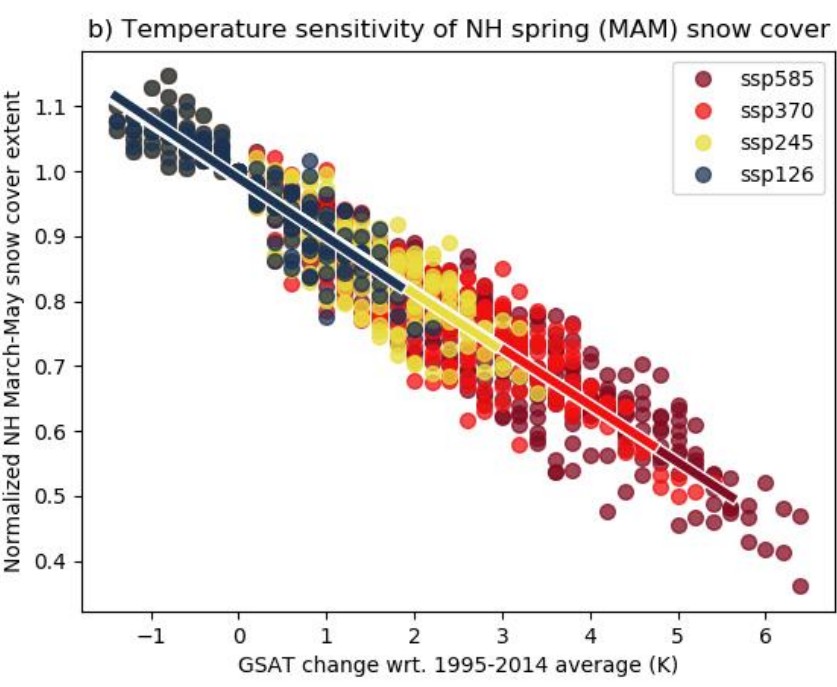


**Figure 9: Spring (March to May) NH snow cover extent against GSAT (relative to the 1995-2014 average) for the CMIP-6 Tier 1 scenarios (SSP1-2.6, SSP2-4.5, SSP3-7.0 and SSP5-8.5), with linear regressions. Each data point is the mean for one CMIP-6 simulation (first ensemble member for each available model) in the corresponding temperature bin.**

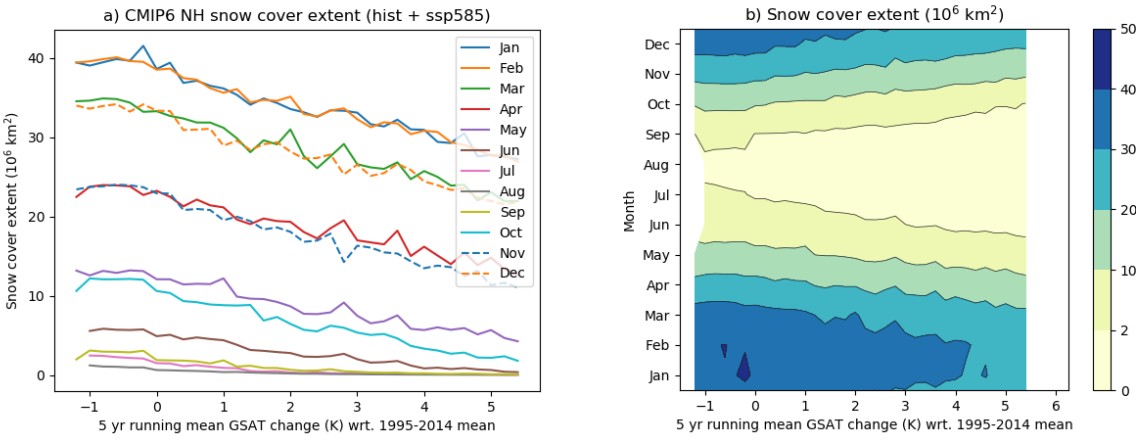

**Figure 10: (a) Multi-model mean CMIP-6 projected Northern Hemisphere snow extent as a function of 5-year mean GSAT for SSP5-8.5. (b) Contour plot showing evolution of monthly snow extent as a function of 5-year mean GSAT.**



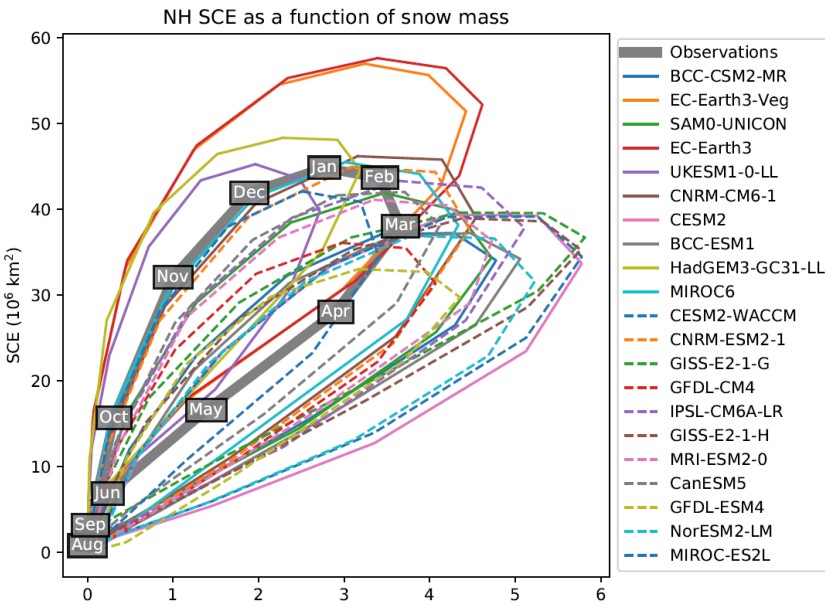


**Figure 11: Relationship between the 1981-2014 monthly mean hemispheric snow cover extent and monthly mean total hemispheric snow mass in the observation-based ensemble-mean and the CMIP-6 models. Summer months are located near (0/0) and data run clockwise through the annual cycle, that is, autumn is on the upper and spring is on the lower branch of the curves.**





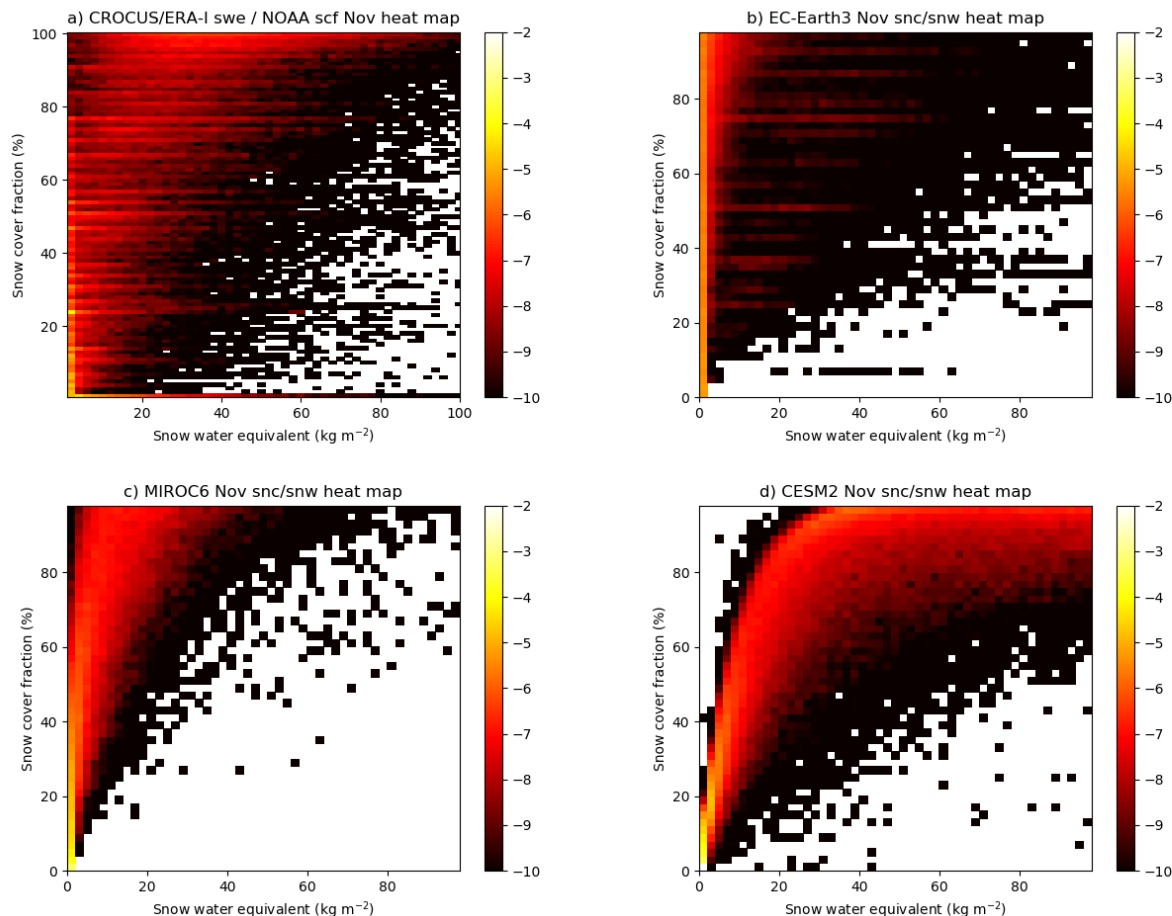


**Figure 12: Relationship between autumn (November) snow mass and snow cover fraction at 1° resolution for CROCUS/ERA-I snow mass vs NOAA snow cover (a, for 1995-2014) and on model grid scale for some selected CMIP-6 models (b-d, 1981-2014). The color shading indicates the relative density of grid points on an arbitrary logarithmic scale.**
