# Peer review of "Historical Northern Hemisphere snow cover trends and projected changes in the CMIP6 multi-model ensemble"

_The Cryosphere, 2019_

## Referee Comment (RC1) · Anonymous Referee #1 · 11 Feb 2020

General comment:

This is a relevant article, presenting significant results based on very recent data from the CMIP-6 project. The manuscript is well-written and pleasant to read. I have no major concerns, only a few suggestions for minor revisions, outlined below.

Specific comments:

- A few things appear throughout the manuscript, showing that it was written by different authors. One of those is the use of capital letters (or absence thereof) for "Northern Hemisphere". Another one is the use of "autumn" or "fall". Please be consistent throughout the manuscript.

[Figure]

- l. 35: "due in part to the insulating properties of snow". This last part of the sentence is a bit confusing and I can't see the causal link with the beginning of the sentence. Could you please elaborate on this?

- l. 88: the acronyms "SCE" and "SWE" are never defined.

- l. 199: There seems to be a sentence missing before the last sentence of the paragraph to explain why the authors refer to "The remaining Tier 1 CMIP-6 ScenarioMIP simulations".

- l. 202: I would maybe change the section title to something like "Observation-based historical trends". As is, the difference between sections 3.1 and 3.2 is not so clear.

- l. 247 and 252: the figure numbering seems wrong. For l. 247, I guess it should read "Figures 5d and 5e" and for l. 252, "Figures 5a and 5b"? Also at l. 254, it would be useful to include: "... in the Hindu-Kush area further west (Figure 5h).

- l. 269: Figure 6 doesn't really show closer agreement between CMIP-6 and observations than for CMIP-5 during the entire fall. The improvement is valid for November and December, but for October for instance, this is not the case. Please reformulate and/or clearly define at some point the seasons you're referring to.

- l. 390: Figure 13d doesn't exist; please replace it by "Figure 12d".

- Figure 2: why does the shading stop in 1967? It would be interesting to see both the grey and blue shadings superimposed during their common period.

- Figures 3 and 4: it is generally not recommended to use both green and red colours on the same figure, especially when they overlap. Please try to find a more colour-blind friendly alternative.

- Figure 5: The resolution and/or quality of this figure seems rather poor. Please improve it.

- Figure 10: I am not sure both panels in this figure are needed, as they show almost

the same thing. I would recommend keeping panel b), and maybe include it in Figure 9.

Technical corrections:

- l. 12 and 13: "An ensemble ... is used" and "a subset ... is used".

- l. 98 and throughout the manuscript: please order your citations by either chronological order or alphabetical order.

- l. 144: please delete the word "for".

- l. 239: "the upper part ... features".

- l. 365: duplicate use of the word "successive". I would replace one of those by the word "consecutive" for example.

- Figure 9: There is an unnecessary "b)" in the figure title.

- Figure 11: the x-axis label is missing. Additionally, in the figure caption, maybe include "... hemispheric snow cover extent (SCE)".

---

## Referee Comment (RC2) · Anonymous Referee #2 · 19 Feb 2020

Overall I thought this was a good paper and will make a good addition to the literature, especially since snow is a critical aspect of the natural environment (with obvious links to transportation, recreation, water supply, etc.) and is projected to decline with continued anthropogenic climate change. I have only minor comments, as enumerated below.

Line 77: Poor spatial resolution likely plays a key role over mountainous areas but is not explicitly mentioned here.

Line 108: Please note somewhere, for example in Table 1, the resolution of the gridded snow mass data sets before regridding to 0.5 degree resolution.

[Figure]

Line 112: Please provide information that would enable the reader to understand to what extent this weighting/mixing was performed. For example, what percent of active grid cells had a GlobSnow weight of 0.5 or less?

Line 117: 4 mm seems somewhat arbitrary. Are your results sensitive to using a different value? For example, does figure 3 only look good because 4 mm was used?

Line 120: You lost me in this part about merging the data sets, not tactically (what you did) but strategically (why you did it). The initial motivation for using multiple data sets was that any one data set is uncertain, and by using multiple data sets you take the uncertainties in observational estimates into account. But now you are removing some of the differences between the data sets, which seems to obviate the motivation for using multiple data sets in the first place. I suggest you augment this section with an explanation of why you chose to take this approach given that one of your motivations was to look at the spread of results across observational data sets, and that this methodology seems to diminish that spread.

Line 148: Off the top of my head, I would have said that the *fractional* variability had likely not changed, but (along with the diminished snow cover in the recent period already seen) the actual variability might have. I.e., if you have more snow overall I'd expect first order that you'd have more variability, but perhaps the same fractional variability. Maybe it's just wording, but it would be good to be clear on this point. Presumably the actual amount of variability scales with the amount of snow, but the fractional variability would be more constant.

Line 161: Although you're not wrong about the emissions following RCP 8.5 from 2006-2014 (mostly, but dropping below that towards the end), in fairness, the global mean temperature between, say, RCP 4.5 and RCP 8.5 are statistically indistinguishable during this period.

Line 164: Typo, should say "SSP3-7.0", not "SSP3-3.7".

Line 166: Regarding "r1i1p1f1", note that the CMIP6 gateway has a warning right at the top of the page that states "WARNING: Not all models include a variant "r1i1p1f1", and across models, identical values of variant_label do not imply identical variants! To learn which forcing datasets were used in each variant, please check modeling group publications and documentation provided through ES-DOC". If that is true, I'm not sure that it is necessarily correct to compare r1i1p1f1's across models. I suppose as the community gains more experience with CMIP6 this will sort itself out. I assume all the models you analyzed did, in fact, have an r1i1p1f1 member?

Line 177: Would I be correct in presuming that models calculate their own snow cover fraction by applying a threshold to their SWE? If so, is there any reason to believe different models used the same threshold? If this is the case and different models used different thresholds, might it have affected your results by adding an element of inconsistency between the analyses of different models?

Line 183: The wording does not make clear whether you used the CNRM-CM6-1 mask for all models, or just the subset of models that did not have the needed masks available. Please clarify. Additionally, if the CNRM mask was only used for a subset of models, please indicate somewhere (in text or in table 2 perhaps) which models used the CNRM mask instead of their own.

Line 191: "IPSL CICLAD computer center" is not sufficiently descriptive. Please augment the acronyms with more specificity, such as spelling it out, location, and institution.

Line 261: "there is clear improvement in mean model agreement with observations at the hemispheric scale..." I presume this is based on Fig. 3, not figure 4 (which shows a degradation in model agreement with observations) or Fig. 5 (which does not show observations at all)? If so, please be a little more thorough on line 261 by noting that it is snow cover extent that shows clear improvement while snow mass shows less agreement with observations.

Line 308: Does each dot in figure 9 represent one year averaged across all models, or

what? Please add a bit more clarity to the text or caption.

Line 371: Please add an X axis label to Fig. 11.

Throughout: I suggest using "autumn" rather than "fall", since "snowfall" is a different thing and too similar to "fall snow".

---

## Author Comment (AC1) · 23 Apr 2020

Thanks to both reviewers for their constructive comments on the manuscript. Changes prompted by these comments have improved the clarity and impact of the paper. Please note that we found and corrected an error in the selection of model simulations analyzed for future projections which affected Figures 8, 9, and 10. Previous figures were being generated using only a single model (which had metrics close to the ensemble median); the updated figures are based on all available models. None of the conclusions were affected, however we performed some additional analysis related to the seasonality of snow sensitivity (Figures 9 and 10) which was incorporated

as additional commentary as shown in the tracked-changes version of the manuscript. Our response to each comment is outlined below in **bold**. Revised text is in *red italics.*

General comment:

This is a relevant article, presenting significant results based on very recent data from the CMIP-6 project. The manuscript is well-written and pleasant to read. I have no major concerns, only a few suggestions for minor revisions, outlined below.

Specific comments:

- A few things appear throughout the manuscript, showing that it was written by different authors. One of those is the use of capital letters (or absence thereof) for "Northern Hemisphere". Another one is the use of "autumn" or "fall". Please be consistent throughout the manuscript.

**We have changed all instances of northern hemisphere to Northern Hemisphere and those of fall to autumn.**

- l. 35: "due in part to the insulating properties of snow". This last part of the sentence is a bit confusing and I can't see the causal link with the beginning of the sentence. Could you please elaborate on this?

**This paragraph was revised as follows:**

*Snow cover influences the carbon balance across biomes and seasons. Across tundra regions in winter, snow cover insulation of the underlying soil is a key factor in driving winter season carbon losses from northern permafrost (Natali et al., 2019). Across the boreal forest in spring, gross primary production and carbon uptake during the subsequent months is directly related to the timing of spring snow melt such that earlier snow melt drives greater carbon uptake (Pulliainen et al., 2017). The net effect of these processes on large scale carbon budgets remains uncertain.*

- l. 88: the acronyms "SCE" and "SWE" are never defined.

**The acronyms were only used in the titles for Sections 2 and 3. We've replaced them with "snow extent" and "snow mass"**

- l. 199: There seems to be a sentence missing before the last sentence of the paragraph to explain why the authors refer to "The remaining Tier 1 CMIP-6 ScenarioMIP simulations".

**We have rephrased the end of the paragraph to read:**

*For analysis of model projections we do not examine simulated trends. Hence the multi-model ensembles analyzed are based on the first realization available for each model.*

- l. 202: I would maybe change the section title to something like "Observation-based historical trends". As is, the difference between sections 3.1 and 3.2 is not so clear **Section 3.1 title changed to:** *"Observation-based trends of snow extent and snow mass"*

**Corresponding title in Discussion (section 4.1) changed to:** *"Observation-based trends"*

.- l. 247 and 252: the figure numbering seems wrong. For l. 247, I guess it should read"Figures 5d and 5e" and for l. 252, "Figures 5a and 5b"? Also at l. 254, it would beuseful to include: "... in the Hindu-Kush area further west (Figure 5h).

**Changed to "Figures 5d-f" at Line 247, "Figures 5a-c" at line 252 and "(Figure 5h)" added at Line 254.**

- l. 269: Figure 6 doesn't really show closer agreement between CMIP-6 and observations than for CMIP-5 during the entire fall. The improvement is valid for November and December, but for October for instance, this is not the case. Please reformulate and/or clearly define at some point the seasons you're referring to.

**This section was rewritten to emphasize the differences between CMIP5 and CMIP6 but now remains agnostic about whether either ensemble has better agreement.**

- l. 390: Figure 13d doesn't exist; please replace it by "Figure 12d".

**Done. Thanks!**

- Figure 2: why does the shading stop in 1967? It would be interesting to see both the grey and blue shadings superimposed during their common period.

**We've provided a new figure for the manuscript with extended shading (Attached Fig 1)**

- Figures 3 and 4: it is generally not recommended to use both green and red colours on the same figure, especially when they overlap. Please try to find a more colour-blind friendly alternative.

**New figures provided (red was changed to blue).**

- Figure 5: The resolution and/or quality of this figure seems rather poor. Please improve it.

**Once finalized, PDF and PS versions of the figures will be submitted.**

- Figure 10: I am not sure both panels in this figure are needed, as they show almost the same thing. I would recommend keeping panel b), and maybe include it in Figure 9.

**We have changed this figure to keep a modified version of panel a to show that the linearity demonstrated in Figure 9 applies more generally for most calendar months. As stated at the beginning of this document, the original version was erroneously presenting results from a single model. The new version shows results from the multi-model ensemble. The kink in the time series at 3.5K (relative to 1995-2014) is due to changes in the number of models available as this level of**

**warming corresponds roughly to the end of the 21st century, beyond which fewer modelling groups contributed realizations. We comment on this in the revised text and demonstrate this explicitly in attached Fig 2.**

Technical corrections: - l. 98 and throughout the manuscript: please order your citations by either chronological order or alphabetical order.

**We have reordered them using chronological order.**

- l. 12 and 13: "An ensemble ... is used" and "a subset ... is used".

- l. 144: please delete the word "for".

- l. 239: "the upper part ... features".

- l. 365: duplicate use of the word "successive". I would replace one of those by the word "consecutive" for example.

**All changed. Thanks!**

- Figure 9: There is an unnecessary "b)" in the figure title. **Removed.**

- Figure 11: the x-axis label is missing. Additionally, in the figure caption, maybe include"... hemispheric snow cover extent (SCE)".

**Changed.**

———————————————

[Figure]

[Figure]

**Fig. 1.** New Figure 2

[Figure]

[Figure]

**Fig. 2.** New Figure 10

[Figure]

**Fig. 3.** Reduced number of models available after +3.5K warming

---

## Author Comment (AC2) · 24 Apr 2020

Thanks to both reviewers for their constructive comments on the manuscript. Changes prompted by these comments have improved the clarity and impact of the paper. Please note that we found and corrected an error in the selection of model simulations analyzed for future projections which affected Figures 8, 9, and 10. Previous figures were being generated using only a single model (which had metrics close to the ensemble median); the updated figures are based on all available models. None of the conclusions were affected, however we performed some additional analysis related to the seasonality of snow sensitivity (Figures 9 and 10) which was incorporated

as additional commentary as shown in the tracked-changes version of the manuscript. Our response to each comment is outlined below in **bold**. Revised text is in *red italics.*

General Comments:

Overall I thought this was a good paper and will make a good addition to the literature, especially since snow is a critical aspect of the natural environment (with obvious links to transportation, recreation, water supply, etc.) and is projected to decline with continued anthropogenic climate change. I have only minor comments, as enumerated below.

Line 77: Poor spatial resolution likely plays a key role over mountainous areas but is not explicitly mentioned here.

**Good point. We added:**

*Despite increases in model resolution since CMIP-3, even the highest resolution models in CMIP-6 are still expected to have challenges simulating snow in mountain regions without appropriate downscaling (Verfaillie et al., 2018).*

Line 108: Please note somewhere, for example in Table 1, the resolution of the gridded snow mass data sets before regridding to 0.5 degree resolution.

**Resolution information was added to Table 1.**

Line 112: Please provide information that would enable the reader to understand to what extent this weighting/mixing was performed. For example, what percent of active grid cells had a GlobSnow weight of 0.5 or less?

**We added the following line to this section:**

*In midwinter, approximately 25% of snow-covered grid cells are blended with less than 50% GlobSnow.*

Line 117: 4 mm seems somewhat arbitrary. Are your results sensitive to using a differ-

ent value? For example, does figure 3 only look good because 4 mm was used?

**The snow extent climatologies of the component datasets do depend on the choice of SWE threshold, however only their variability is incorporated into the mean time series. The climatology is that of the NOAA/Rutgers dataset, which we assume to be more accurate than the reanalyses and GlobSnow. This is in contrast to the mean snow mass time series, for which no preferred climatology exists. The snow extent anomalies of the component data sets are also affected to a smaller extent by the choice of SWE threshold, however the difference between a 4mm and a 10mm threshold affects the mean trends to less than about 5%. Threshold choices less than 2mm will begin to affect things more substantially and are probably unrealistic (see Figure 10, for example, from the following reference):** *Krinner et al., ESM-SnowMIP: assessing snow models and quantifying snow-related climate feedbacks, Geosci. Model Dev., 11, 5027–5049, 2018.*

Line 120: You lost me in this part about merging the data sets, not tactically (what you did) but strategically (why you did it). The initial motivation for using multiple datasets was that any one data set is uncertain, and by using multiple data sets you take the uncertainties in observational estimates into account. But now you are removing some of the differences between the data sets, which seems to obviate the motivation for using multiple data sets in the first place. I suggest you augment this section with an explanation of why you chose to take this approach given that one of your motivations was to look at the spread of results across observational data sets, and that this methodology seems to diminish that spread.

**Thanks for bringing this up! We have added the following additional text to Section 2.1 which hopefuly provides a more complete rationale for the method:**

*The ideal dataset for comparison with coupled model ensembles would be a single, bias-free, observational record with uncertainty due only to interannual variability. Analysis of such a record would yield an accurate trend with a simply interpretable un-*

*certainty due to record length and sampling uncertainty and would provide the most straightforward way to compare the observed trend with those from coupled model ensembles (which contain further spread due to model structure and parametrization as well and internal variability). In reality, observational records are not ideal and different datasets may contain additional uncertainty in their variability due to technical issues. Our goal here is to combine anomalies from an ensemble of observation-based estimates in order to mitigate the most prominent trend biases of the individual datasets and estimate the irreducible interannual variability.*

**We also note here that because the focus of this paper was on evaluating CMIP6, we have refrained from a more thorough discussion of the full uncertainty/spread in trends among component gridded datasets. However we plan to present such an analysis in a subsequent contribution.**

Line 148: Off the top of my head, I would have said that the *fractional* variability had likely not changed, but (along with the diminished snow cover in the recent period already seen) the actual variability might have. I.e., if you have more snow overall I'd expect first order that you'd have more variability, but perhaps the same fractional variability. Maybe it's just wording, but it would be good to be clear on this point. Presumably the actual amount of variability scales with the amount of snow, but the fractional variability would be more constant.

**We have rephrased the original text, which was misleading. The standard deviation of the Brown index (as well as its climatology) is actually sampled over 1972-1991 and it is this value that is scaled to the ensemble average standard deviation sampled from the 1981-2014 period. Hence the two periods used are reasonably close and differences in the choice of period are covered by the additional uncertainty used to plot the blue shading (furthermore, most of this additional uncertainty is related to the choice of sample period used to match the NOAA climatology, rather than the sampling period for the standard deviation). The paragraph now reads:**

*For certain months, the Northern Hemisphere snow extent time series can be extended back to 1922 through the interpolation of a fixed network of surface snow depth observations, as described in Brown (2000). These observations have sufficient spatial coverage to estimate relative changes in snow extent from year to year in the form of an index, but are not estimates of the actual snow extent. Limitations in the mutual availability and coverage of surface observations over North America and Eurasia mean that hemispheric estimates are only available for March, and April. March and April indices are converted into units of snow extent separately over each continent using the same process described above. Each index is standardized based on the1972-1991 period. Then the index climatology is matched to that from NOAA sampled over the same period. We chose the 1972-1991 reference period because we expect that the NOAA climatology over 1972-1991 is more representative of earlier 20th century snow cover (whereas snow cover will have already responded to warming temperatures by the latter portion of the record). We rescale the variability of each index using the 6-member ensemble mean standard deviation for the appropriate month and continent that was used to compute adjusted time series for the other data sets. This process has an implicit assumption that the variability sampled in the Brown index over 1972-1991 is comparable to that from the ensemble mean SCE time series over the 1981-2014 period. The rescaled indices for each continent are then summed together to obtain a historical estimate of Northern Hemisphere snow cover extent from 1922-1991. Uncertainty in the Brown time series is calculated using the standard error of forecast, however we consider additional uncertainty due to the choice of reference periods used for matching the climatology and standard deviation. 95% uncertainty bounds are calculated for all possible selections of sequential 20 year climatological periods from the NOAA record in combination with all possible sequential 20 year estimates of variability. The maximum and minimum uncertainty bounds calculated from these combinations are used to define the range for each year.*

Line 161: Although you're not wrong about the emissions following RCP 8.5 from 2006-2014 (mostly, but dropping below that towards the end), in fairness, the global mean

temperature between, say, RCP 4.5 and RCP 8.5 are statistically indistinguishable during this period.

**We have rephrased the sentence to say observed emissions 'reasonably follow' those of RCP8.5.**

Line 164: Typo, should say "SSP3-7.0", not "SSP3-3.7". **Changed.**

Line 166: Regarding "r1i1p1f1", note that the CMIP6 gateway has a warning right at the top of the page that states "WARNING: Not all models include a variant "r1i1p1f1" and across models, identical values of variant label do not imply identical variants! To learn which forcing datasets were used in each variant, please check modeling group publications and documentation provided through ES-DOC". If that is true, I'm not sure that it is necessarily correct to compare r1i1p1f1's across models. I suppose as the community gains more experience with CMIP6 this will sort itself out. I assume all the models you analyzed did, in fact, have an r1i1p1f1 member?

**Sampling differences of climatological snow extent and snow mass across different realizations can be expected to be small. Specifying r1i1p1/r1i1p1f1 is a simple way to choose a single realization from each model keeping in mind that some groups only submit a single model. For trends the sampling differences will be much larger, which is why the median trend is selected from available model realizations as stated in the text. We have also added the following additional specification to the text:**

*"A handful of models had no r1i1p1f1 variant. In these cases the r1i1p1f2 variant was chosen (e.g. the CNRM models, MIROC-ESM, UKESM) and barring that the r1i1p1f3 variant (HadGEM). NorESM realization r1i1p1f1 had 10 years of missing data and was ignored for trends; realization r2i1p1f1 was used for its climatology."*

Line 177: Would I be correct in presuming that models calculate their own snow cover fraction by applying a threshold to their SWE? If so, is there any reason to believe

different models used the same threshold? If this is the case and different models used different thresholds, might it have affected your results by adding an element of inconsistency between the analyses of different models?

**Yes, however this is a genuine parametrization choice made by different modeling center and bears scrutiny as examined in the Discussion Section.**

Line 183: The wording does not make clear whether you used the CNRM-CM6-1 mask for all models, or just the subset of models that did not have the needed masks available. Please clarify. Additionally, if the CNRM mask was only used for a subset of models, please indicate somewhere (in text or in table 2 perhaps) which models used the CNRM mask instead of their own.

**The CNRM masks were used throughout for consistency. We've rephrased the sentence to read:**

*We therefore used the masks from the CNRM-CM6-1 coupled model (Voldoire et al., 2019) for all other models by interpolating these masks to the corresponding grid of each model.*

Line 191: "IPSL CICLAD computer center" is not sufficiently descriptive. Please augment the acronyms with more specificity, such as spelling it out, location, and institution.
. **Acronyms specified.**

Line 261: "there is clear improvement in mean model agreement with observations at the hemispheric scale..." I presume this is based on Fig. 3, not figure 4 (which shows a degradation in model agreement with observations) or Fig. 5 (which does not show observations at all)? If so, please be a little more thorough on line 261 by noting that it is snow cover extent that shows clear improvement while snow mass shows less agreement with observations.

**Fair point. This has been changed to:**

*In summary, while some regional differences in SCF are evident, the improved hemi-*

*spheric statistics for snow extent seen in Figure 3 are generally consistent with the geographic patterns shown in Figure 5.*

**Please note, however that observations are shown in Fig 5 with pink contour.**

Line 308: Does each dot in figure 9 represent one year averaged across all models, or what? Please add a bit more clarity to the text or caption. **We have changed the part of the caption to the following:**

*Each data point represents the average snow extent from one CMIP-6 simulation (first ensemble member for each available model). The average is taken over all pentads from the given simulation which fall into a given temperature bin.*

Line 371: Please add an X axis label to Fig. 11.

**Changed.**

Throughout: I suggest using "autumn" rather than "fall", since "snowfall" is a different thing and too similar to "fall snow".

**Changed.**